# Post-Consumer Poly(ethylene terephthalate) (PET) Depolymerization by *Yarrowia lipolytica*: A Comparison between Hydrolysis Using Cell-Free Enzymatic Extracts and Microbial Submerged Cultivation

**DOI:** 10.3390/molecules27217502

**Published:** 2022-11-03

**Authors:** Julio Cesar Soares Sales, Aline Machado de Castro, Bernardo Dias Ribeiro, Maria Alice Zarur Coelho

**Affiliations:** 1Departamento de Bioquímica, Instituto de Química, Universidade Federal do Rio de Janeiro, Av. Athos da Silveira Ramos, 149. Ilha do Fundão, Rio de Janeiro 21941-909, Brazil; 2Divisão de Biotecnologia, Centro de Pesquisa e Desenvolvimento, PETROBRAS, Av. Horácio Macedo, 950. Ilha do Fundão, Rio de Janeiro 21941-915, Brazil; 3Departamento de Engenharia Bioquímica, Escola de Química, Universidade Federal do Rio de Janeiro, Av. Athos da Silveira Ramos, 149. Ilha do Fundão, Rio de Janeiro 21941-909, Brazil

**Keywords:** *Yarrowia lipolytica*, lipase, plastics, Biocatalysis

## Abstract

Several microorganisms have been reported as capable of acting on poly(ethylene terephthalate) (PET) to some extent, such as *Yarrowia lipolytica*, which is a yeast known to produce various hydrolases of industrial interest. The present work aims to evaluate PET depolymerization by *Y. lipolytica* using two different strategies. In the first one, biocatalysts were produced during solid-state fermentation (SSF-YL), extracted and subsequently used for the hydrolysis of PET and bis(2-hydroxyethyl terephthalate) (BHET), a key intermediate in PET hydrolysis. Biocatalysts were able to act on BHET, yielding terephthalic acid (TPA) (131.31 µmol L^−1^), and on PET, leading to a TPA concentration of 42.80 µmol L^−1^ after 168 h. In the second strategy, PET depolymerization was evaluated during submerged cultivations of *Y. lipolytica* using four different culture media, and the use of YT medium ((*w*/*v*) yeast extract 1%, tryptone 2%) yielded the highest TPA concentration after 96 h (65.40 µmol L^−1^). A final TPA concentration of 94.3 µmol L^−1^ was obtained on a scale-up in benchtop bioreactors using YT medium. The conversion obtained in bioreactors was 121% higher than in systems with SSF-YL. The results of the present work suggest a relevant role of *Y. lipolytica* cells in the depolymerization process.

## 1. Introduction

The world production of plastics continues to increase year by year; until 2017, it was estimated that the production reached 8300 million metric tons, and by 2050, 12,000 megatons (Mt) will accumulate in the environment and landfills [1]. Serious environmental problems are caused by plastic accumulation in terrestrial and marine ecosystems, which incur microplastic formation [2]. The increase in marine pollution was also strongly impacted by the COVID-19 pandemic, as it required the use of various personal protective equipment made from plastic resins [3]. Bondaroff & Cooke (2020) [4] pointed to an entry of 1.56 billion masks into the oceans in 2020, resulting in an accumulation of another 6240 tons in the marine system. It is estimated that about 5 trillion pieces of plastic are in the oceans, which is equivalent to approximately 270 tons [5].

In an attempt to mitigate the volume of waste generated, several recycling processes have been studied and implemented on an industrial scale, such as mechanical, chemical recycling and even pyrolysis [6]. However, of all plastic waste generated worldwide between 1950 and 2015, only 9% was recycled and 60% was disposed of in landfills or the environment [1]. Considering the various plastic resins currently produced (e.g., polyethylene, polypropylene and poly(ethylene terephthalate) (PET)), biotechnology presents a promising route to deal with post-consumer waste [7]. This route is based on biodegradation using microorganisms and/or microbial enzymes in order to obtain monomers or other molecules of interest with greater added values [8,9]. Regarding post-consumer plastics biodegradation technologies, PET is the polymer that presents the greatest maturities in relation to the proposed technologies [10].

Since the first PET hydrolase, a cutinase, was reported in 2005, several microbial enzymes have been reported for PET biodepolymerization to obtain its monomers (e.g., lipases, esterases, cutinases and *p*-nitrobenzyl esterases) [11,12,13,14,15]. In 2016, Yoshida et al. [16] reported the isolation of a bacterium from plastic waste capable of producing an enzyme with a high affinity for PET, defined as PETase (EC 3.1.1.101). Other microorganisms have been reported to be able to act during PET depolymerization, including *Yarrowia lipolytica* [17,18]. *Y. lipolytica* is a nonconventional yeast widely studied for its intense secretory activity and ability to produce several biomolecules of industrial interest (e.g., organic acids, lipases, esterases and proteases) [19,20]. One of the major products of *Y. lipolytica* are lipases; the production of these enzymes by the yeast is reported either by submerged cultures or by solid-state fermentation (SSF) [19,21]. One of the great appeals of the SSF, which is characterized by the absence or near absence of free water in the system, is linked to the possibility of using residues and/or by-products from the agroindustry as a cultivation medium to obtain various biomolecules, directly impacting the associated costs to the process [22,23].

In this context, the present work aims to evaluate the application of enzymes produced by *Y. lipolytica* in SSF in the biodepolymerization of PET and compare it with depolymerization by submerged cultures containing *Y. lipolytica* cells and its secreted enzymes, aiming to understand the dynamics of the process and draw new perspectives on the use of this yeast for PET depolymerization.

## 2. Results and Discussion

### 2.1. Hydrolysis Using Cell-Free Enzymatic Extracts

In these hydrolysis assays using cell-free enzymatic extracts, two different substrates were evaluated, PC-PET and BHET, a key intermediate in PET hydrolysis. The biocatalysts used in this step were produced by *Y. lipolytica* through solid-state fermentation (SSF) in media containing soybean bran and post-consumer PET (PC-PET) (20 wt%), as previously defined by Sales et al. [18]. Another enzyme used in the process was Lipase B from *Candida antarctica* (CALB), which is already reported in the literature as an enzyme of interest in the hydrolysis of BHET [24]. Figure 1 presents the product concentration values obtained during PET and BHET hydrolysis.

Based on the results obtained during BHET hydrolysis using CALB (Figure 1a), it is possible to verify the high efficiency of this enzyme in hydrolyzing this substrate, being possible to observe a high conversion after 1 h of reaction. Complete hydrolysis can be found after 24 h of processing, as seen in the molar fractions (χ) presented in Table 1.

These biocatalysts have already been employed by Castro et al. [13] to reverse mono-(hydroxyethyl) terephthalate (MHET) accumulation during a PET depolymerization reaction catalyzed by *Humicola insolens* cutinase. The authors reported complete hydrolysis to TPA of high MHET concentrations (~7000 µmol L^−1^) after 24 h.

When analyzing biocatalysts produced by *Y. lipolytica* in SSF (Figure 1b), it is observed that, after 168 h, there is still a nonhydrolyzed substrate (440 µmol L^−1^), and the MHET concentration increases concomitantly with the decrease in BHET. In other words, the process leads to MHET as the primary hydrolysis product, as shown in the molar fraction values in Table 1. The TPA concentration remains unchanged between 1 h and 168 h. A similar profile of BHET biodegradation was observed by Qiu et al. (2020) using an isolated strain of Enterobacter sp. (HX1). The authors reported that the microorganism could produce an esterase that acts during the hydrolysis of BHET; however, it is first hydrolyzed to MHET and not directly to TPA. In addition, they reported a half-life of BHET (500 mg L^−1^) of approximately 60 h. The present work presents a concentration below half of the initial one already in 48 h.

The increased MHET concentration throughout the process may be since MHET acts as a possible competitive inhibitor for *Y. lipolytica* enzymes. The inhibition phenomenon is already reported for other biocatalysts used in the enzymatic hydrolysis of PET, such as the cutinase from Thermobifida fusca [25]. One of the alternatives proposed in the literature is the use of multienzymatic systems that allow an efficient hydrolysis of MHET and prevent its accumulation in the reaction medium [24]. Costa et al. [17] observed a very different behavior in submerged cultivations of *Y. lipolytica* IMUFRJ 50682 with the addition of BHET (500 mg L^−1^). Although the authors reported MHET as the major product, the total consumption of BHET was observed close to 48 h, which may be related to cellular metabolism. The consumption of this substrate may be associated with previous extracellular hydrolysis and the entry of the products into cellular metabolism or with a prior uptake followed by hydrolysis catalyzed by intracellular enzymes.

When analyzing biocatalysts performance in the hydrolysis of PET-PC, it is observed that the catalysis performed by CALB leads to the release of a single aromatic product, TPA. This may be due to the high specificity of CALB in converting BHET and MHET intermediates into TPA. The results agree with those pointed out by Castro et al. (2017) [13], who showed very low values of TPA and MHET during PC-PET hydrolysis reactions catalyzed by CALB. From Figure 1d, it is possible to analyze the performance of biocatalysts produced by *Y. lipolytica* when PC-PET was used as a substrate. Although both systems show low PET to TPA conversions, using the pool of enzymes produced by *Y. lipolytica* led to higher concentrations of hydrolysis products at the end of the reaction. Table 2 presents the conversion and molar fraction values of the PC-PET assays.

The TPA concentration using SSF-YL after 168 h is almost three times higher when compared to assays catalyzed by CALB. In addition to TPA, MHET and BHET were also formed during the reaction. However, TPA was the primary product from 24 h onwards and remained until the end, presenting a molar fraction of 0.618 in 168 h. Processes that lead to high molar fractions of monomers are interesting for the technology downstream; such molecules can be recovered and used directly in the polymerization process.

### 2.2. PC-PET Depolymerization in Submerged Cultivations

#### 2.2.1. Depolymerization in Flasks

In addition to evaluating the depolymerization of PC-PET using enzymes produced by *Y. lipolytica* in SSF, depolymerization tests by submerged fermentation were also carried out. From these assays, which, in addition to enzymes, contain *Y. lipolytica* cells, it was possible to investigate the role of the microorganism in the process. For this, four media with different compositions (YP, YT, YPD and YTD) were chosen to evaluate the impact of the media components on yeast metabolism during the process. Figure 2 shows the growth profiles in the different assays.

From the cell growth data presented in Figure 2, accelerated growth can be observed, resulting in high cell concentrations within 20 h of the process. This may result from an increase in mass transfer associated with such cultures caused by a decreased working volume, commonly from 200 mL to 100 mL in 500-mL flasks. In addition, similar behaviors can be inferred for pairs of medium with and without glucose. Those without sugar, a readily assimilable carbon source, showed a drop after 18 h of cultivation. This may be linked to a depletion of nutrients and a low synthesis of reserves, such as glycogen, in such media. Bhutada et al. [26] evaluated glycogen synthesis by *Y. lipolytica* under different nutritional stress conditions. The authors reported glycogen synthesis is carried out in the exponential growth phase and a greater synthesis in conditions where the limitation is tied to nitrogen and not carbon. Synthesis under such conditions was almost three times higher than under glucose shortages. The drop in cell concentration may also be associated with cell adhesion to the polymer surface, leading to an underestimation of the actual values of cell quantification. In a previous work (unpublished data), a SEM analysis showed the adhesion of *Y. lipolytica* cells to the polymer surface.

Quite different behavior can be observed in cultures with the presence of glucose. After 18 h, the cell concentration increases and begins to stabilize around 48 h. From *Y. lipolytica* growth data in the different media, it was possible to determine the growth parameters, such as the specific growth rate (µ), generation time (t_g_), maximum cell concentration (X_max_) and biomass productivity (Q_x_). These parameters are presented in Table 3.

As can be seen in Table 3, the highest values of µ are observed in media containing glucose. As the µ values are lower in media without glucose, the generation times are longer for these conditions. These data are in accordance with what was observed by Costa et al. [17] during *Y. lipolytica* IMUFRJ 50682 cultures targeting the degradation of PC-PET in the presence and absence of glucose. However, the behavior had only been evaluated in media containing peptone as a nitrogen source.

The biodegradation process was also monitored to produce several enzymes throughout the cultures. Figure 3 presents the values of lipolytic activity during the 96 h of the assay.

From Figure 3, it can be inferred that lipase production by *Y. lipolytica* was higher in media that did not contain glucose. This is possibly related to the repression of sugar in the production of lipases. Higher phosphorylation activities directly impact the expression of the Lip2 gene [27]. Furthermore, it is observed that the nitrogen source plays a fundamental role in the production of lipases by *Y. lipolytica*. Both sources (peptone and tryptone) are widely reported in the literature as components of the lipase production medium. However, Fickers et al. [28] reported tryptone as a more adequate source for the production of this enzyme, which is confirmed in the present work. In fact, the high production of lipases in the YTD medium, which has glucose and high phosphorylation activity, was probably modulated by the presence of tryptone. It is essential to mention that, as well as a sustained cell concentration observed, this medium also showed the production of lipases up to 48 h of culture, being the only medium to present lipolytic activity after 24 h. Despite the intense secretion of lipases in the YTD medium, the highest productivity of the assay was observed in the YT medium, which reached 115 U L^−1^ h^−1^ in 6 h of cultivation.

In addition to verifying the production of lipases, the esterase activity of the medium was also quantified. The production of esterases over 96 h of cultivation is shown in Figure 4.

When observing the esterase production throughout the culture, it can be seen that, unlike lipases, these enzymes are produced throughout the culture in the four evaluated media. This may be associated with an inductive effect of the polymer and its ester bonds. The reports on the production of esterase (Lip7) from *Y. lipolytica* in submerged cultures are related to the use of tryptone as the main source of nitrogen [29,30]. High productivities were obtained in the media without a readily assimilable carbon source, the highest being in the tryptone medium (YT): 286 U L^−1^ h^−1^ in 4 h of the process. The intense secretion of enzymes may be linked to the possible attempt to use PET as a carbon source by cells. In fact, a significant increase in esterase activity was observed simultaneously as the beginning of the decline in cell concentration due to possible nutrient depletion.

In addition to lipases and esterases, the production of proteases throughout the process was also evaluated. Figure 5 shows the values of proteolytic activity over the cultivation.

When analyzing the values of proteolytic activity, it is observed that the media that presented carbon source limitations showed an intense secretion of proteases from the beginning. This may be related to the possible assimilation of proteolysis products by *Y. lipolytica* as a carbon source. Interestingly, the medium that presented the lowest production of proteases during the tests, YTD, was the medium during which the most prolonged lipase production was observed. Therefore, the enzymes showed greater stability associated with lower proteolytic activity. Another important observation was that, even with high proteolytic activity since the beginning of the culture, the media without glucose maintained esterase activity throughout the process. The phenomenon may be associated with the fact that *Y. lipolytica* esterase is a glycosylated enzyme, similar to Lip2, which confers greater resistance to proteolysis. Furthermore, Lip7 is a cell-bound enzyme, and as it is naturally immobilized in the cell wall, there is possible greater stability in the presence of proteases [29].

As a result of the metabolic activity of *Y. lipolytica*, an alkalinization of pH is observed throughout the PET depolymerization process presented in Figure 6. A strong parallel can be established with the proteolytic activity, since the increase is more discreet in the media with less activity, such as the YPD and YTD. The process is intensified under conditions where there is no addition of glucose and is a result of the high release of amine groups during proteolysis.

To verify the yield of the biodepolymerization process and the products generated, the supernatant was evaluated for its content of PET hydrolysis products. In the tests shown in Figure 7, where molar concentrations of hydrolysis products are shown, it is observed that the media containing tryptone and no glucose added led to the highest TPA concentration value after 96 h: 65.4 µmol L^−1^. This is probably a result of the high enzymatic activities obtained throughout the cultivation. In fact, a steep increase in the concentration of TPA was observed during the initial phase of cultivation, where there was intense secretion of both esterases and lipases, which were only produced for up to 20 h in this medium. From the analysis of Figure 7a,b, it seems that the cultivation in YT medium led to a concentration of TPA almost 20% higher than the same process conducted with YP, the second best, after 96 h. In addition to TPA, the other hydrolysis products also showed higher concentrations simultaneously.

TPA concentrations obtained in the YT trials were almost five times higher than those reported by Costa et al. [17] in the PET-PC biodepolymerization assays using *Y. lipolytica* IMUFRJ 50682 through submerged cultivation. The reason for the increase is probably related to the higher enzymatic activity observed in the media of the present work when compared to the media used by the authors: YP. In addition, smaller particle sizes were used (0.15–0.50 mm), which resulted in a larger contact surface, facilitating the biodepolymerization process. The investigation of the particle size effect on PET depolymerization reactions was also carried out by Castro et al. [31] using cutinase from H. insolens. The authors reported an increase from 4.2 to 20.3 mM/day in the TPA release rate when decreasing the particle size from values greater than 1.4 mm to values smaller than 0.21 mm.

A parallel can be drawn between the TPA concentrations obtained in the submerged assays with YT medium and in the assay using the biocatalysts produced in SSF (SSF-YL). At 96 h, the concentration of TPA in the YT medium was two times higher than that obtained with the biocatalysts produced in SSF. The conversion obtained at 96 h (2.52%) was 53% higher than that obtained using SSF-YL at 168 h. This may reveal that the microorganism plays a key role in biodepolymerization. It is worth remembering that two mechanisms of assimilation of hydrophobic compounds are reported for *Y. lipolytica* [32]: mediated transport, which consists of the production of biosurfactants; and direct transport, which consists of cell adhesion to the substrate. Therefore, for an effective biodepolymerization by enzymes, some other factors seem to be relevant. To verify this, it is necessary to investigate cell adhesion to the plastic, the possible production of biosurfactants and other proteins that may be involved in the process.

Media with glucose (YPD and YTD) showed lower TPA concentration values than those without sugar. Even the YTD medium showed higher enzymatic activities than the YP; for example, these values were lower after 96 h. It is important to point out that the yeast may have consumed part of the hydrolysis products generated as a carbon source, as seen in the drop in TPA concentration between 18 and 20 h in Figure 7b. Costa et al. (2020) reported the consumption of TPA by *Y. lipolytica* in a rich medium containing yeast extract (1% *w*/*v*) and peptone (2% *w*/*v*).

In addition to the HPLC analyses, the ATR-FTIR analysis was performed to verify the changes in the material after treatment in different media. Figure 8 presents the spectra obtained during the analyses.

The ATR-FTIR spectra presented in Figure 8 show characteristic bands of PET, such as the one near 1730 cm^−1^ (C=O stretch), the one near 1240 cm^−1^ (C-C-O stretch) and the one near 1040 cm^−1^ (O-C-C stretch). Concerning biological treatment, an increase in transmittance in these regions can be noted in all conditions evaluated compared to PET without treatment. This effect is desired, since ester bonds are cleaved in the biodepolymerization process.

Ioakeimidis et al. (2016) [33] evaluated the biodegradation of PET in marine environments and adopted FTIR as the process monitoring technique. The authors report decreases in absorbance in regions related to the ester group (C=O, C-C-O and O-C-C stretch) and also in the area close to 730 cm^−1^. This region is related to the C-H bond in the aromatic domain. It is essential to mention that changes in this region were also observed in all conditions evaluated in the present work, with the highest increase in transmittance observed in the YTD medium (~9%). In this condition, it was also possible to observe the highest increases related to C=O and C-C-O stretching: 7 and 8%, respectively.

Sammon et al. (2000) [34], in a study evaluating the hydrolysis of PET films using KOH (1%), showed changes in the FTIR spectrum very similar to the present work. The authors also pointed out an increase in absorption in the region between 2800 and 3000 cm^−1^ related to the aliphatic domain’s C–H bonds. This is probably due to the decrease in the polymer chain size, which results in a greater number of ethylene glycol terminals. A reduction of transmittance values in this region was observed in biological treatments with *Y. lipolytica*, which may evidence a decrease in chain size caused by the biodepolymerization process. The results obtained in ATR-FTIR were in line with those observed in the other analyses, where PET hydrolysis products and hydrolytic enzymes were observed during the cultivation. Interestingly, the medium that presented the most significant changes in FTIR (YTD) was not the same, showing the highest concentrations of hydrolysis products after 96 h. This may be related to the consumption of these molecules by *Y. lipolytica* to maintain the cell metabolism.

#### 2.2.2. Depolymerization in Bioreactors

From the results obtained in the biodepolymerization assays in the flasks, the YT condition was chosen to be evaluated on a larger scale, in benchtop bioreactors. This condition was determined, because it presented the highest values of enzymatic activity and also the highest conversion values. The YT medium in the bioreactors showed a similar growth profile to that observed in the flasks, reaching its highest cell concentration at 20 h (10.3 g L^−1^). The cell concentration and pH data obtained in the bioreactor are shown in Figure 9.

There was a lag phase during the first two hours of cultivation, followed by the beginning of the exponential. The µ value obtained for the cultivation was lower than that observed in the shaken flasks, 0.150 h^−1^, which may reflect the low agitation of the system. After 20 h of cultivation, a decrease in cell concentration was also noticed, which may be related to the adhesion of the cells to the polymer surface, leading to a drop in the optical density recorded. The cell adhesion of *Y. lipolytica* IMUFRJ 50682 on the surface of PET has already been reported in the literature [35]. The cell adhesion to the polymer may be due to the high hydrophobicity of the cell wall of this strain of *Y. lipolytica* [36]. In addition, the drop in optical density observed may be linked to cell death due to nutrient depletion. Regarding the pH values presented, the same pattern of alkalinization of the medium, as compared to the investigations in shake flasks, was observed over 96 h due to the metabolic activity.

Quantification of the enzymatic activities during the cultivation was also performed in these assays, since the biodepolymerization was associated with the secretion of hydrolases by the yeast. The enzymatic activities obtained during the 96-h process are shown in Figure 10.

As seen in Figure 10, lipase production was already during the first 6 h of cultivation. The absence of a readily assimilable carbon source may be related to this early lipase production. In other studies, the production of lipase by *Y. lipolytica* IMUFRJ 50682 was associated with the almost complete depletion of glucose, used as a carbon source in the culture medium [37]. The production of lipases was not sustained after 18 h of cultivation, where there was no detection of lipolytic activity in the medium, as could already be observed in the flask assays.

Esterase production occurs after two hours of cultivation and is maintained for 96 h. The main esterase of *Y. lipolytica* (Lip7) is an enzyme adhered to the cell wall during all growth phases, so its activity detected in the supernatant tends to be underestimated. The maintenance of esterase activity throughout the cultivation, by a possible induction by ester bonds in the polymer structure, is of great importance for the continuity of the biodepolymerization process and the increase in the concentration of hydrolysis products.

In addition to the presence of lipases and esterases, the production of proteases is also observed after 6 h of cultivation, which intensifies after 18 h. The intensification of protease production may be associated with a depletion of readily assimilable nitrogen sources and the absence of carbon sources. Furthermore, the secretion of proteases by *Y. lipolytica* is associated with pH. It can be seen that the pH values presented in Figure 9 are all above 6.5 after two hours of cultivation, and alkaline protease secretion is induced at values above 6.5 [38]. The proteolytic activity may have affected the activity of other enzymes present in the medium.

The investigation of PET hydrolysis products was performed by HPLC, and the concentrations are shown in Figure 11.

From Figure 11, it can be seen that the enzymes produced by *Y. lipolytica* were able to act in the PET biodepolymerization process, as observed in the preliminary tests. There was an increase in the concentration of the PET aromatic monomer terephthalic acid, reaching a concentration above 94 µmol L^−1^ in 96 h. Other important information that can be extracted from Figure 11 is that TPA remains the primary product during cultivation. The higher TPA concentration values led to an increase in the obtained conversion of almost 44% when compared to the results obtained in the shake flask assays. This may be related to an even greater decrease in particle size caused by the shearing phenomenon caused by the bioreactor impellers, thus increasing the contact surface. The reduction in particle size caused by the action of the impellers was observed by Kadic et al. [39] during the hydrolysis of sugarcane bagasse by cellulases in benchtop bioreactors. The authors reported a significant increase in the presence of smaller particles concomitant with the increase in agitation, which went from 100 rpm to 300 rpm.

MHET, the hydrolysis intermediate, was detected at low concentrations (~2 mg/L), and BHET was not detected. The quantification of the metabolites associated with *Y. lipolytica* was also performed in a refractive index detector, where it was possible to observe the presence of trehalose in the culture medium. The production of trehalose, a compatible solute, by *Y. lipolytica* during the process may be associated with nutritional scarcity during cultivation.

To evaluate the biodepolymerization process and consolidate the results obtained by HPLC, other analyses were performed on the material after the biological treatment, such as ATR-FTIR. Figure 12 presents the results obtained by this technique.

From the ATR-FTIR analysis shown in Figure 12, it is observed, as in the tests in the flasks, that the same profile of bands are characteristic of PET (close to 1730 cm^−1^, at 1240 cm^−1^, at 1040 cm^−1^ and 730 cm^−1^) [33,40]. Regarding the impact after cultivation in a bioreactor, the same profile of modifications in the polymer can be noted. The increase in transmittance in the regions close to 1730 cm^−1^, 1240 cm^−1^, 1040 cm^−1^ and 730 cm^−1^ were 4, 5, 5 and 6%, respectively.

## 3. Materials and Methods

### 3.1. Materials

Soybean bran was purchased from Caramuru Alimentos (Itumbiara, Goiás, Brazil). Post-consumer PET (PC-PET) was obtained from a plastic industrial recycling plant located in Rio de Janeiro (Brazil) and gently provided by PETROBRAS (Rio de Janeiro, Brazil). Materials were ground in a knives mill (TE-648, Tecnal), and only particles below 1.18 mm were used in the cultivations.

Terephthalic acid (TPA) and bis(2-hydroxyethyl) terephthalate (BHET) were purchased from Sigma-Aldrich. Mono-(hydroxyethyl) terephthalate (MHET) was obtained from the hydrolysis of BHET using *Humicola insolens* cutinase [24].

Lipase B from *Candida antarctica* was purchased from Sigma-Aldrich.

### 3.2. Microorganism

A wild strain of *Yarrowia lipolytica* (IMUFRJ 50682) isolated from an estuary in Guanabara Bay (Rio de Janeiro, Brazil) [41] was used. Cells were stocked at 4 °C in YPD agar medium ((*w*/*v*) 1% yeast extract, 2% peptone, 2% dextrose and 3% agar). Prior to the fermentation process, cells were grown in YPD (Yeast extract–Peptone–Dextrose) medium ((*w*/*v*) 1% yeast extract, 2% peptone and 2% dextrose) for 72 h in a rotary shaker at 28 °C and 160 rpm.

### 3.3. Solid-State Fermentation (SSF) and Hydrolysis Using Cell-Free Enzymatic Extracts

SSF was conducted in tray-type reactors containing 10 g of substrate. As a substrate, a mixture of soybean bran and PC-PET (20 wt%) was used, as previously defined by Sales et al. [18]. Moisture was adjusted using an emulsion of water and soybean oil (1.5 wt%) once optimized by Souza et al. [42]. The medium was inoculated with 0.71 mg cells/g substrate, as previously set by Farias et al. [43]. Bioreactors were incubated for 14 h in a Biochemical Oxygen Demand (BOD) Incubator Chamber (TECNAL, Piracicaba, Brazil) at 28 °C under saturated moisture (99%), as set by Souza et al. [42].

Biocatalysts recovery was made using phosphate buffer (50 mM) at pH 7.0. Fifty milliliters of buffer were added to the fermented solid and then placed in a rotary shaker at 37 °C and 200 rpm for 20 min. Samples were manually pressed with gauze to remove larger particles and then centrifuged at 3000 rpm for 5 min to remove cells and smaller particles. The crude enzymatic extract was stored at −20 °C for later analysis.

Post-consumer PET (PC-PET) and BHET, a key intermediate in PET hydrolysis, were used as the substrates for enzymatic hydrolysis. Biocatalysts produced through SSF (SSF-YL) and *Candida antarctica* Lipase B (CALB) were used in these reactions. Assays were conducted in 10-mL Erlenmeyer flasks, with 4 mL of working volume, containing 46 U of esterase activity at 37 °C and 180 rpm for 168 h.

### 3.4. Submerged Cultivation in Flasks (SCF) and Submerged Cultivation in Bioreactors (SCB)

SCF was conducted in 500-mL Erlenmeyer flasks using 100 mL of medium. Four different media were tested in biodepolymerization tests: YP ((*w*/*v*) 1% yeast extract, 2% peptone); YPD ((*w*/*v*) 1% yeast extract, 2% peptone and 2% dextrose); YT ((*w*/*v*) 1% yeast extract, 2% tryptone) and YTD ((*w*/*v*) 1% yeast extract, 2% tryptone and 2% dextrose) containing 0.5 g L^−1^ of PC-PET in each one. Initial cell concentrations were set for 1 g L^−1^, and the assays were incubated in a rotary shaker at 28 °C and 250 rpm for 96 h.

SCB was conducted in 0.5-L bioreactors (Multifors, INFORS HT) equipped with two Rushton impellers containing 0.45 L of YT medium ((*w*/*v*) 1% yeast extract, 2% tryptone) and 0.5 g L^−1^ of PC-PET. Assay conditions were set for 28 °C, 450 rpm and 1 vessel volumes per minute (vvm).

### 3.5. Analytical Methods

#### 3.5.1. Biomass Quantification, pH and Growth Parameters

Biomass quantification in submerged cultivations was done in a spectrophotometer at 570 nm, and the optical density values were converted in g of dry weight cell L^−1^ using a factor obtained through a dry weight cell curve. pH values were obtained using a bench pH meter (TECNAL, Piracicaba, Brazil).

Specific growth rate (µ) was obtained by the slope of the exponential growth portion in the semi-log plot, and the maximum cell concentration (X_max_) was the highest cell concentration observed during cultivation. Biomass productivity (Q_x_) was obtained by dividing the cell concentration by the time, and the generation time (*t_g_*) was obtained using Equation (1).
(1)$$tg=\frac{ln2}{µ}$$

#### 3.5.2. Enzymes Activities

Lipase activity was determined by spectrophotometry through the analysis of the absorbance variation during the hydrolysis of *p*-nitrophenyl laurate (*p*-NPL) (0.56 mM) to *p*-nitrophenol (*p*-NP) at 410 nm, as reported by Pereira-Meirelles et al. [44]. One enzyme unit (U) was defined as the amount of enzyme capable of catalyzing the release of 1 μmol of *p*-nitrophenol per minute under the assay conditions.

Esterase activity was determined by spectrophotometry through the analysis of the absorbance variation during the hydrolysis of *p*-nitrophenyl butyrate (*p*-NPB) (2.5 mM) to *p*-nitrophenol (*p*-NP) at 410 nm, as reported by Carniel et al. [24]. One enzyme unit (U) was defined as the amount of enzyme capable of catalyzing the release of 1 μmol of *p*-nitrophenol per minute under the assay conditions.

Protease activity was determined by an adapted methodology of the protocol proposed by Charney & Tomarelli [45]. Azocasein was used as a substrate for protease activity determination. The protocol consisted of azocasein hydrolysis for 40 min, followed by the precipitation of nonhydrolyzed proteins using trichloroacetic acid (TCA) (15% *w*/*v*) and centrifugation at 3000 rpm for 15 min. Absorbance evaluation was done on a microplate reader (SpectraMax m2e, Molecular Devices) at 428 nm by filling the microplate with 100 µL of KOH 6 N and the later addition of 100 µL of the supernatant. One enzyme unit (U) was defined as the amount of enzyme capable of promoting an increase of 0.01 unit in absorbance per minute under the reaction conditions.

#### 3.5.3. PET Hydrolysis Products Quantification

The PET hydrolysis products (TPA, BHET and MHET) were quantified using High-Performance Liquid Chromatography (HPLC) in a Shimadzu Nexera LC-40 (Shimadzu, Kyoto, Japan). It used an Agilent Eclipse Plus C18 (5 µm, 4.6 × 250 mm) column, a UV detector at 254 nm and an analysis temperature of 30 °C. A gradient mixture of acetonitrile and formic acid (0.05%) was used as a mobile phase with a flow rate of 0.5 mL min^−1^. The injection volume was set to 10 µL.

Mole fraction evaluation of the samples (*χ_i_*, where *i* = *TPA*, *MHET* and *BHET*) was calculated using Equation (2). This equation considers the moles ratio of each component (*n*) and the sum of the moles of these three components.
(2)$$\chi_{i}=\frac{n_{i}}{n_{TPA}+n_{MHET}+n_{BHET}}$$

#### 3.5.4. ATR-FTIR (Attenuated Total Reflectance—Fourier-Transform Infrared Spectroscopy)

The ATR-FTIR spectra of the polymers before and after the biodepolymerization process were obtained with the Shimadzu spectrometer model IRTracer-100 (Shimadzu, Japan). Spectra were acquired between 400 and 4100 cm^−1^ at a resolution of 4 cm^−1^.

## 4. Conclusions

The results obtained in the present work revealed that the yeast *Y. lipolytica* proved to be a potential microorganism to act in the biodepolymerization of PET. The application of enzymes produced in SSF could act in the enzymatic hydrolysis of BHET and PC-PET during 168 h of hydrolysis.

Biodepolymerization assays in submerged cultures of *Y. lipolytica* in four different media effectively converted PC-PET. The media containing tryptone (YT and YTD) showed the intense secretion of hydrolytic enzymes, such as lipases and esterases. The highest concentration of TPA after 96 h was that of YT medium: 65.4 µmol L^−1^. This condition was scaled to the benchtop bioreactors, and a 121% higher concentration of TPA was observed after 96 h: 94.31 µmol L^−1^.

From the results presented in this work, it is clear that submerged systems are better for the biodepolymerization of PC-PET by *Y. lipolytica*. It is still necessary to investigate the possible mechanisms associated with the process, such as the production of biosurfactants, proteins and the cell adhesion profiles in different media. Another essential factor to be understood is the possible steps involved in the assimilation of PET monomers. In addition, strategies to maintain the enzymatic activity and cell concentration throughout the process should be investigated, as well as better media formulations to minimize the associated costs.

## Figures and Tables

**Figure 1 molecules-27-07502-f001:**
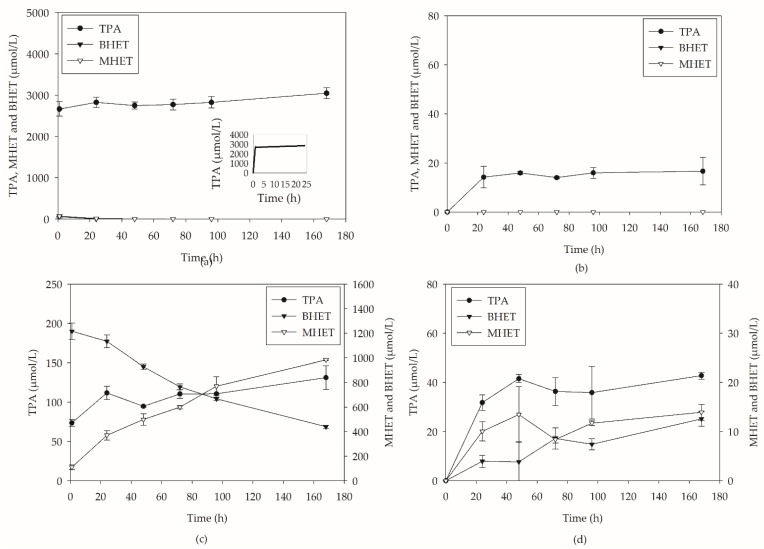
Enzymatic hydrolysis of BHET using: (**a**) CALB and (**b**) SSF-YL and PC-PET using: (**c**) CALB and (**d**) SSF-YL at 37 °C and 180 rpm.

**Figure 2 molecules-27-07502-f002:**
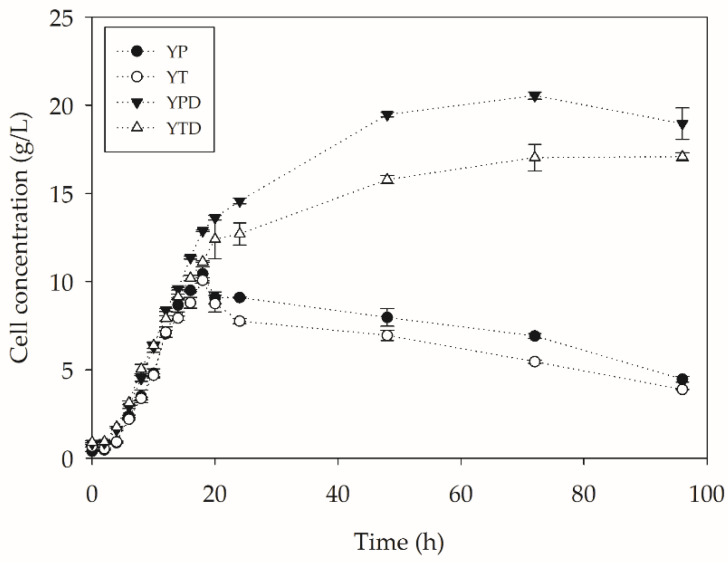
Growth of *Y. lipolytica* IMUFRJ 50682 in different media containing PET-PC (500 mg/L) at 28 °C and 250 rpm for 96 h.

**Figure 3 molecules-27-07502-f003:**
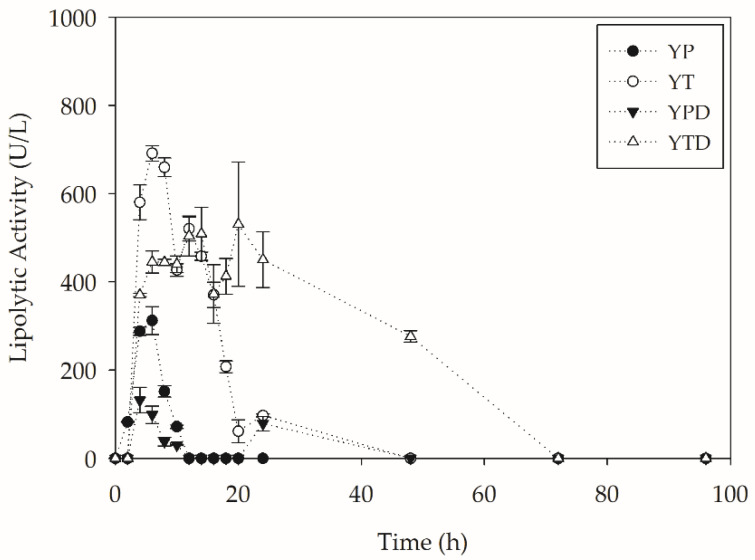
Lipolytic activity during PC-PET biodepolymerization by submerged cultures of *Y. lipolytica* IMUFRJ 50682 at 28 °C and 250 rpm for 96 h.

**Figure 4 molecules-27-07502-f004:**
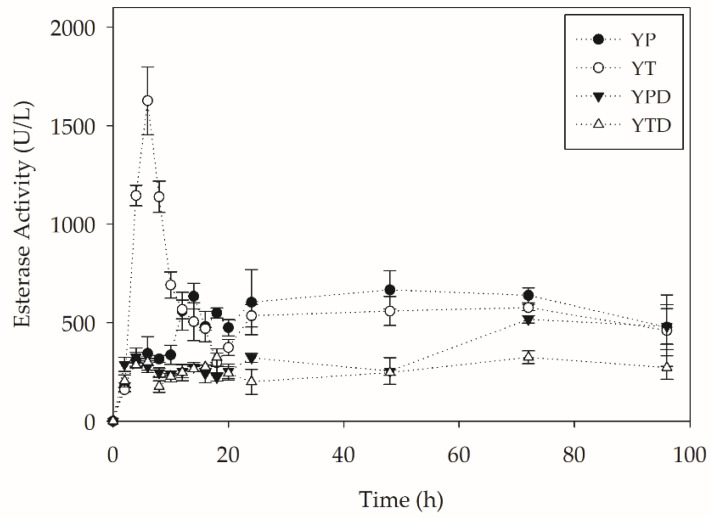
Esterase activity during PC-PET biodepolymerization by submerged cultures of *Y. lipolytica* IMUFRJ 50682 at 28 °C and 250 rpm for 96 h.

**Figure 5 molecules-27-07502-f005:**
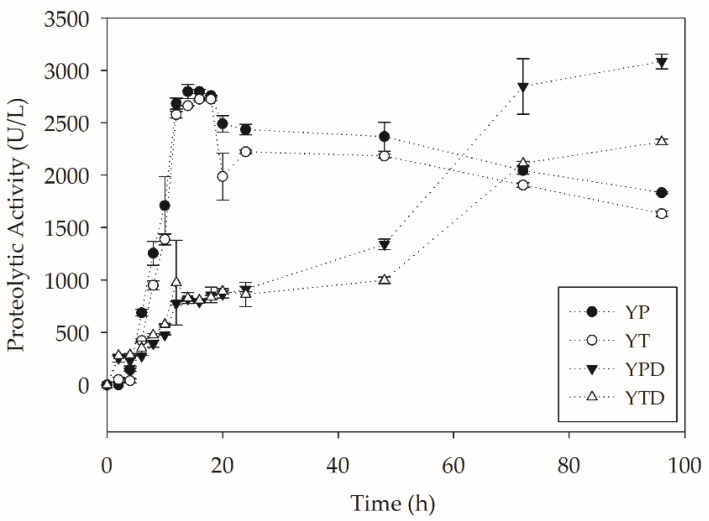
Protease activity during PC-PET biodepolymerization by submerged cultures of *Y. lipolytica* IMUFRJ 50682 at 28 °C and 250 rpm for 96 h.

**Figure 6 molecules-27-07502-f006:**
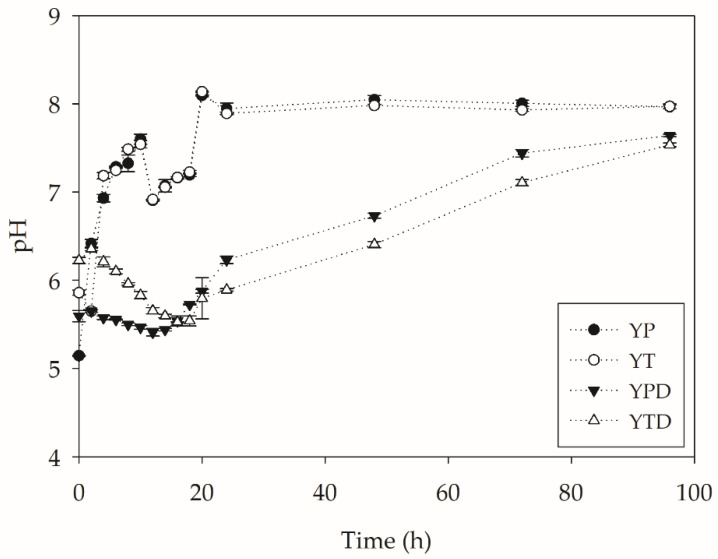
pH variations during PC-PET biodepolymerization by submerged cultures of *Y. lipolytica* IMUFRJ 50682 at 28 °C and 250 rpm for 96 h.

**Figure 7 molecules-27-07502-f007:**
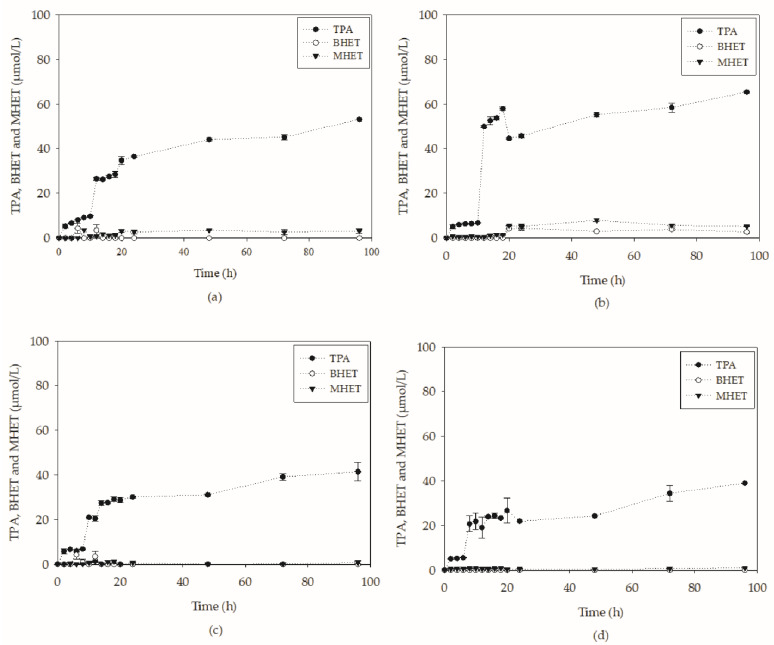
Hydrolysis products of PC-PET obtained during submerged cultivation of *Y. lipolytica* IMUFRJ 50682 in (**a**) YP, (**b**) YT, (**c**) YPD and (**d**) YTD medium at 28 °C and 250 rpm for 96 h.

**Figure 8 molecules-27-07502-f008:**
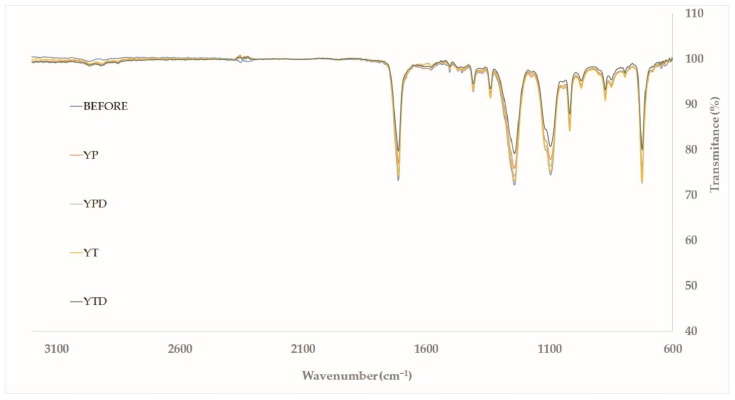
ATR-FTIR spectrum before and after biodepolymerization during submerged cultures of *Y. lipolytica* IMUFRJ 50682 at 28 °C and 250 rpm for 96 h.

**Figure 9 molecules-27-07502-f009:**
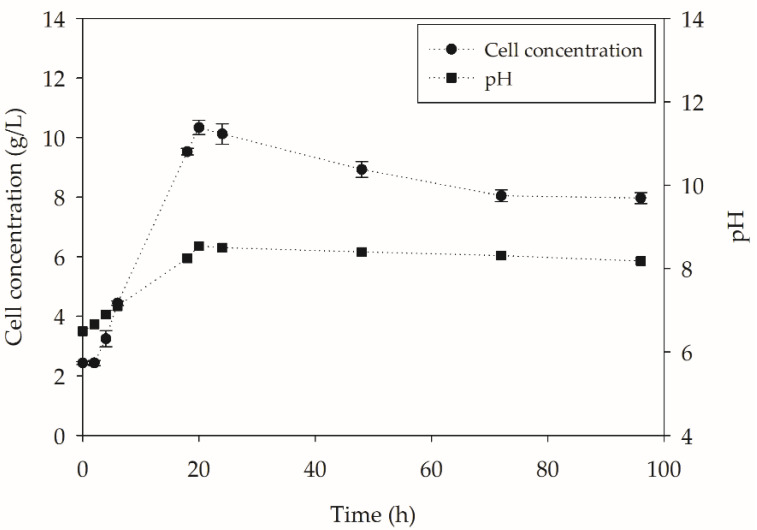
*Y. lipolytica* IMUFRJ 50682 growth and pH variation in the bioreactors (450 rpm, 1 vvm and 28 °C) containing YT medium + PC-PET (500 mg/L) over 96 h.

**Figure 10 molecules-27-07502-f010:**
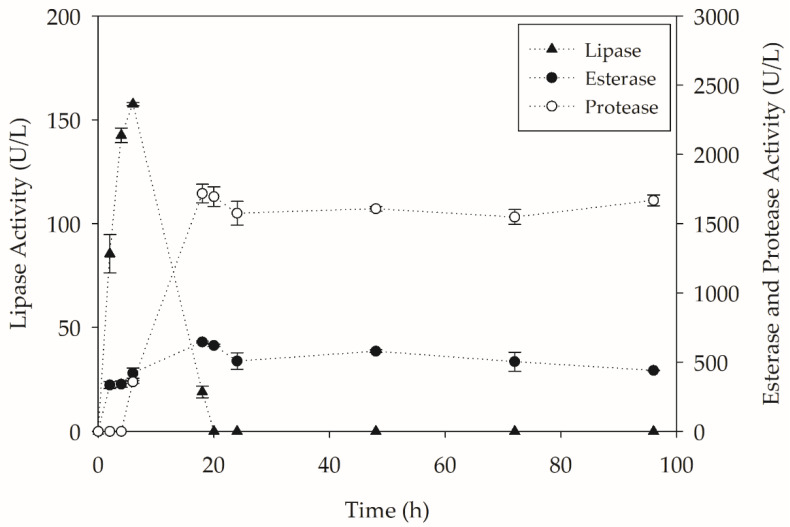
Enzymes production by *Y. lipolytica* IMUFRJ 50682 in bioreactors (450 rpm, 1 vvm and 28 °C) containing YT medium + PC-PET (500 mg/L) over 96 h.

**Figure 11 molecules-27-07502-f011:**
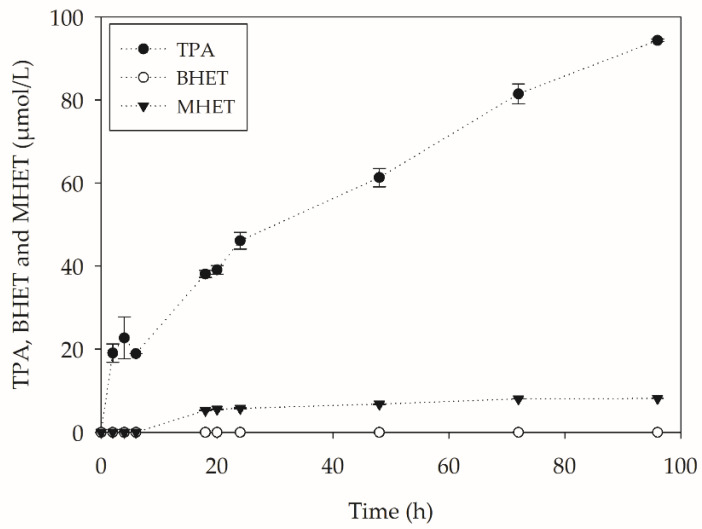
PET hydrolysis products obtained during *Y. lipolytica* cultivation in bioreactors (450 rpm, 1 vvm and 28 °C) containing YT medium + PC-PET (500 mg/L) for 96 h.

**Figure 12 molecules-27-07502-f012:**
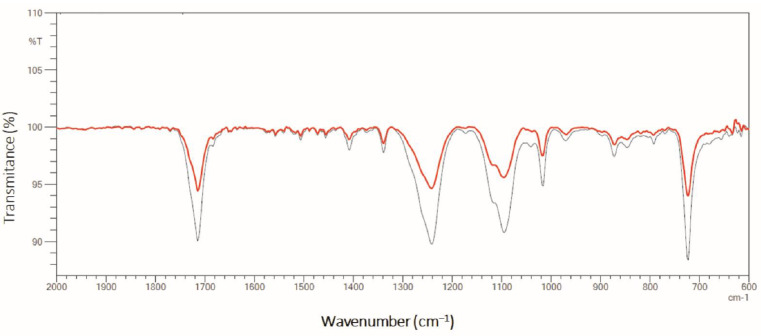
ATR-FTIR spectrum before (black line) and after (red line) the biodepolymerization process during *Y. lipolytica* cultivation in bioreactors (450 rpm, 1 vvm and 28 °C) containing YT + PC-PET (500 mg/L) for 96 h.

**Table 1 molecules-27-07502-t001:** Molar fractions after 168 h of reaction using BHET and PC-PET as hydrolysis substrates at 37 °C and 180 rpm.

Assay	χ_tpa_	χ_bhet_	χ_mhet_
CALB + BHET	1	0	0
SSF-YL + BHET	0.084	0.284	0.632

**Table 2 molecules-27-07502-t002:** Conversion and molar fractions after 168 h of reaction using PC-PET as the hydrolysis substrates at 37 °C and 180 rpm.

Assay	χ_tpa_	χ_bhet_	χ_mhet_	Conversion (%)
CALB + PC-PET	1	0	0	0.65
SSF-YL + PC-PET	0.618	0.182	0.201	1.64

**Table 3 molecules-27-07502-t003:** Growth parameters of *Y. lipolytica* IMUFRJ 50682 in different media at 28 °C and 250 rpm.

Assay	µ (h^−1^)	t_g_ (h)	X_max_ (g L^−1^)	Q_x_ (g L^−1^ h^−1^)
YP	0.271	2.56	10.45	0.580
YT	0.262	2.65	10.08	0.560
YPD	0.302	2.29	20.54	0.716
YTD	0.304	2.28	17.03	0.618

## Data Availability

Not applicable.

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
