# Peer review of "Post-Consumer Poly(ethylene terephthalate) (PET) Depolymerization by Yarrowia lipolytica: A Comparison between Hydrolysis Using Cell-Free Enzymatic Extracts and Microbial Submerged Cultivation"

_molecules, 2022, doi:10.3390/molecules27217502_

Round 1

Reviewer 1 Report

Abstract:

Please clarify the type of microorganism. There are few terms that need to be extend; SSF-YL and YT.  

L26: Why cell?

Keywords: must be different from the title. 

Introduction:

L32-33: Why metric? 1200 of what? Expand Mt

L37: ref

1.56 unit? 

Line 37-40: The sentence is very confusing, please revise.

Line 52-54: Confusing sentence needs revision.

Line 60 and Line 62: The sentence patterns are similar.

It is important to highlight the molecules rather than the process, if you chose Molecules Journal.

Result and discussion:

2.1 Have this cell-free enzyme been characterised previously? How do you describe the increase in production of the substates?

Table 1: How did you obtain the result after 24 hr from this tab'e?

L110-112: It is interesting to report the half-life. 

Line 113-122: propose how can you mitigate this problem? 

Line 126: Ref

Fig 2. Unit?  and please check throughout the manuscript

Line 167: Missing reference to backup

Table 3: Explain each terms in the table 

*I would combine the data in the small number of tables and I have to admit that I lose track from time to time reading this manuscript. 

Line 375: Missing discussion along with the stament in L382

3. Materials and methods;

Since the abstract sections the work into two, should not be wise to rearrange MM accordingly?

Expand YPD

Line 439: ..., and 3% agar

Line 445: what are the ration of these substrates? Should it be substrate mixture of...? 

Line 447: 0.71 mg cells/g substate with what cultures?

Line 449: Expand BOD and add country of manufacturing.

Line 450: ^at pH

Line 453: How can we be sure that all cell has been removed at this speed?

Line 459-461: Missing reference thereof.

Line468: Unit.

Line 474: Expand vvm

Line 480: missing Brand of pH meter

Line: 505: Missing country of manufacturing (same goes for 3.7.4

Author Response

Rio de Janeiro, October 19th, 2022.

Thank you for your e-mail dated 10th October 2022 regarding the article entitled “Post-consumer poly(ethylene terephthalate) (PET) depolymerization by Yarrowia lipolytica: A comparison between hydrolysis using cell-free enzymatic extracts and microbial submerged cultivation” (molecules-1950263) and for reviewer’s comments.

Kindly find enclosed a revised version taking into consideration that reviewer’s comments have been very seriously considered and answered. Note: the new changes are included in red in the revised manuscript.

Let me close by thanking you in advance for the time spent dealing with this manuscript. 

Yours sincerely,

Julio Cesar Soares Sales, MSc.

Biochemistry Department

Chemistry Institute

Federal University of Rio de Janeiro

Reviewer 1:

Comments and Suggestions for Authors

Abstract:

- Please clarify the type of microorganism. There are few terms that need to be extend; SSF-YL and YT.  

R: We acknowledge reviewer’s consideration and included information about the microorganism in the abstract.

“..such as Yarrowia lipolytica, which is a yeast regarded to produce various hydrolases of industrial interest.”

Regarding the terms present in the abstract, SSF-YL was defined as the biocatalysts produced in solid-state fermentation and the YT medium had its composition also presented. To make it clear that YT is the culture medium, the term medium was added.

“..and the use of YT medium ((w/v) yeast extract 1%, tryptone 2 %)..”

- L26: Why cell?

R: The processes carried out in submerged fermentation have the presence of Yarrowia lipolytica cells. Although the depolymerization of PET is still mediated by enzymes, other factors can also influence the process, such as cell adhesion to the polymer and the production of biosurfactants. Such discussions are present throughout the manuscript.

- Keywords: must be different from the title. 

R: We appreciate the comment and inform you that we have removed one of the keywords.

Introduction:

- L32-33: Why metric? 1200 of what? Expand Mt

R: Metric tons is a unit of mass. We expanded Mt in the manuscript.

“..12000 megatonnes (Mt) will accumulate in the environment and landfills..”

- L37: ref

R: Both the period that ends in line 37 and the one that begins there have references.

- 1.56 unit? 

R: The unit is presented in the sentence. There are 1.56 billion masks.

- Line 37-40: The sentence is very confusing, please revise.

R: We appreciate the reviewer's consideration and inform that we reformulated the sentence.

“Bondaroff & Cooke (2020) [4] pointed to an entry of 1.56 billion masks into the oceans in 2020, resulting in an accumulation of another 6,240 tons in the marine system. It is estimated that about 5 trillion pieces of plastic are in the oceans, which is equivalent to approximately 270 tons [5].”

- Line 52-54: Confusing sentence needs revision.

R: We appreciate the reviewer's consideration and inform you that we have changed the order of the sentence for better clarity.

“Since the first PET hydrolase, a cutinase, was reported in 2005, several microbial enzymes have been reported for PET biodepolymerization to obtain its monomers (e.g., lipases, esterases, cutinases, p-nitrobenzyl esterases)..”

Line 60 and Line 62: The sentence patterns are similar.

R: We acknowledge reviewer’s consideration, but the sentences do not show the same pattern. In the first one, it is discussed that Yarrowia lipolytica biocatalysts can be produced by submerged fermentation and by solid-state fermentation (SSF). In the second, the SSF is conceptualized and its advantages are presented.

It is important to highlight the molecules rather than the process, if you chose Molecules Journal.

R: This work aims to evaluate the performance of Yarrowia lipolytica enzymes in the enzymatic hydrolysis of PET. In the two scenarios evaluated here, the process is mediated by yeast lipases and esterases. For greater clarity, we have added information that enables the reader to understand that enzymes are also present during submerged cultivation.

“by submerged cultures containing Y. lipolytica cells and its secreted enzymes..”

Result and discussion:

- 2.1 Have this cell-free enzyme been characterised previously? How do you describe the increase in production of the substates?

R: As described in the manuscript, this culture medium was defined in previous works of our research group. The article is cited in the text.

In these previous works, the presence of PET in the culture medium proved to be a potential inducer in the production of esterases, enzymes capable of catalyzing the hydrolysis of ester bonds. The enzymes produced in this culture medium also showed higher performance in the enzymatic hydrolysis of PET.

- Table 1: How did you obtain the result after 24 hr from this tab'e?

R: The results were obtained using Equation 1 presented in materials and methods.

- L110-112: It is interesting to report the half-life. 

R: We acknowledge reviewer’s consideration, but we do not have this exact result, however, as presented in the text, values ​​below half of the initial concentration are observed from 48 hours onwards, which is 12 hours lower than that observed by the reference cited in the manuscript.

- Line 113-122: propose how can you mitigate this problem? 

R: We acknowledge reviewer’s consideration and we have included a proposal from the literature to mitigate inhibition by MHET.

“One of the alternatives proposed in the literature is the use of multienzymatic systems that allow an efficient hydrolysis of MHET and prevent its accumulation in the reaction medium [24].”

- Line 126: Ref

R: We included the reference on line 126.

“The results agree with those pointed out by Castro et al. (2017) [25], who showed..”

- Fig 2. Unit?  and please check throughout the manuscript

R: The two axes in Figure 2 present unit. Cell concentration (g/L) and time (h).

- Line 167: Missing reference to backup

R: We understand the consideration, however the sentence itself shows that works not yet published by the group reveal the adhesion of Yarrowia cells to the polymer.

Furthermore, in the discussion of the results of cell growth in the bioreactor, a reference of adhesion of the strain of Yarrowia lipolytica used in the present work on the surface of PET is presented.

“The cell adhesion of Y. lipolytica IMUFRJ 50682 on the surface of PET is already reported in the literature [36].”

- Table 3: Explain each terms in the table

R: We acknowledge reviewer’s consideration and included the terms explanation.

“From Y. lipolytica growth data in the different media, it was possible to determine the growth parameters, such as specific growth rate (µ), generation time (tg), maximum cell concentration (Xmax) and biomass productivity (Qx). These parameters are presented in Table 3.”

*I would combine the data in the small number of tables and I have to admit that I lose track from time to time reading this manuscript.

R: We appreciate the suggestion, however joining the tables is unfeasible. Since they are discussed at different times and, according to the molecules guideline, the table or figure must be presented right after its citation in the text. Another reason is the fact that table 3 presents data completely different from the other two tables.

- Line 375: Missing discussion along with the stament in L382

R: From line 375 to line 378, the importance of the esterase activity observed in the bioreactor throughout the entire cultivation is discussed. The importance of the topic had also been pointed out in the flask assays discussion.

From line 379 onwards, the production of proteases, another enzyme detected in cultivation, is discussed. In addition to the activity, the impact of production is addressed, which is strictly linked to the pH of the medium. The discussion includes important references that support the hypotheses presented in the present work.

  1. Materials and methods;

- Since the abstract sections the work into two, should not be wise to rearrange MM accordingly?

R: We acknowledge reviewer’s consideration and rearranged M&M.

- Expand YPD

R: We acknowledge reviewer’s consideration and expanded YPD.

Line 439: ..., and 3% agar

R: We acknowledge reviewer’s consideration and included this suggestion.

Line 445: what are the ration of these substrates? Should it be substrate mixture of...? 

R: We used a mixture of soybean bran and PC-PET as substrate for solid-state fermentation. As described in M&M, the proportion used was 80% soybean bran to 20% PC-PET.

Line 447: 0.71 mg cells/g substate with what cultures?

R: As described in the M&M, there is only one medium for solid-state fermentation, which is the one containing soybean bran and PC-PET as substrate.

Line 449: Expand BOD and add country of manufacturing.

R: We acknowledge reviewer’s consideration and included this suggestion.

Line 450: ^at pH

R: We acknowledge reviewer’s consideration and included this suggestion.

Line 453: How can we be sure that all cell has been removed at this speed?

R: This is a very common speed for removing yeast cells, being used in several protocols.

Line 459-461: Missing reference thereof.

R: This protocol for PET hydrolysis using SSF-YL and CALB in this conditions was proposed by our team in this work.

Line468: Unit.

R: There is no missing unit in Line 468.

Line 474: Expand vvm

R: We acknowledge reviewer’s consideration and included this suggestion.

Line 480: missing Brand of pH meter

R: We acknowledge reviewer’s consideration and included this suggestion.

Line: 505: Missing country of manufacturing (same goes for 3.7.4

R: We acknowledge reviewer’s consideration and included this suggestion.

Reviewer 2 Report

The manuscript “Post-consumer poly(ethylene terephthalate) (PET) depolymerization by Yarrowia lipolytica: A comparison between hydrolysis using cell-free enzymatic extracts and microbial submerged cultivationis an interesting study within the environmental research community.

The manuscript is well presented, and the information is relevant to the journal. Yet, I have some recommendations which I include here.

Introduction

PPE is not used again in the manuscript; thus, the abbreviation is unnecessary.

Results

MHET is not defined.

Please include how μ, tg, xmax and Qx were obtained.

Materials and methods

3.3

Why use 14 h of SSF? Are there any preliminary reports? Data?

3.4

Was the CALB produced also vía SSF? What were the conditions? Was it used as a control? Please specify.

Is it 46 U per mL?

Were 10 mL of total or working volume?

What was the solid-to-liquid ratio used for the hydrolysis?

3.6

0.45 of 0.5 L for the working volume seems high (90%). Was the headspace enough to support the process conditions (450 rpm, 1 vvm)?

3.7.1 Please specify that biomass quantification was done for the submerged fermentation.

Figures

1.     Is it possible to separate TPA in a secondary axis for a better view?

7. Same as figure 1.

8. Please, improve resolution.

12. Please, improve resolution.

Conclusions

Authors mentioned in the introduction: “One of the great appeals of the SSF, which is characterized by the absence or near absence of free water in the system, is linked to the possibility of using residues and/or by-products from the agroindustry as a cultivation medium…”

Given the results, from the authors’ perspective, is SSF still feasible? could it be improved? is SSF still worthy of study for this type of bioprocess?

References

Please update references (2022).

Author Response

Rio de Janeiro, October 19th, 2022.

Thank you for your e-mail dated 10th October 2022 regarding the article entitled “Post-consumer poly(ethylene terephthalate) (PET) depolymerization by Yarrowia lipolytica: A comparison between hydrolysis using cell-free enzymatic extracts and microbial submerged cultivation” (molecules-1950263) and for reviewer’s comments.

Kindly find enclosed a revised version taking into consideration that reviewer’s comments have been very seriously considered and answered. Note: the new changes are included in red in the revised manuscript.

Let me close by thanking you in advance for the time spent dealing with this manuscript. 

Yours sincerely,

Julio Cesar Soares Sales, MSc.

Biochemistry Department

Chemistry Institute

Federal University of Rio de Janeiro

Introduction

PPE is not used again in the manuscript; thus, the abbreviation is unnecessary.

 R: We acknowledge reviewer’s consideration and removed this abbreviation.

Results

MHET is not defined.

R: We acknowledge reviewer’s consideration and included the definition of MHET in its first appearance.

“These biocatalysts have already been employed by Castro et al. [25] to reverse mono-(hydroxyethyl) terephthalate (MHET) accumulation during PET depolymerization reaction catalyzed by Humicola insolens cutinase.”

- Please include how μ, tg, xmax and Qx were obtained.

R: We acknowledge the suggestion and inform that we included this information in the M&M section.

“Specific growth rate (µ) was obtained by the slope of the exponential growth portion in the semi-log plot and maximum cell concentration (Xmax) was the highest cell concentration observed during cultivation. Biomass productivity (Qx) was obtained by dividing the cell concentration by the time and generation time (tg) was obtained using Equation 2.”

 Materials and methods

3.3

Why use 14 h of SSF? Are there any preliminary reports? Data?

R: This time was set by previous works of our group. It’s now cited in the M&M section.

“Bioreactors were incubated for 14 hours in a Biochemical Oxygen Demand (BOD) Incubator Chamber (TECNAL, Brazil) at 28 °C under saturated moisture (99%) as set by Souza et al [43].”

3.4

Was the CALB produced also vía SSF? What were the conditions? Was it used as a control? Please specify.

R: CALB was purchased from Sigma-Aldrich. We included this information in the M&M section.

“Lipase B from Candida antarctica was purchased from Sigma-Aldrich.”

- Is it 46 U per mL?

R: No. It is just 46 U. Based on the activity of the enzymatic extract (U/L), we calculated this value.

- Were 10 mL of total or working volume?

R: No. The working volume was 4 mL. We included this information in the M&M section.

“Assays were conducted in 10 mL Erlenmeyer flasks, with 4 mL of working volume, containing 46 U of esterase activity at 37 °C and 180 rpm for 168 hours.”

- What was the solid-to-liquid ratio used for the hydrolysis?

R: It was used 500 mg/L of PC-PET. We included this information in the M&M section.

3.6

- 0.45 of 0.5 L for the working volume seems high (90%). Was the headspace enough to support the process conditions (450 rpm, 1 vvm)?

R: Yes. It is the condition adopted in our group for several applications in this model of bioreactor.

- 3.7.1 Please specify that biomass quantification was done for the submerged fermentation.

R: We acknowledge reviewer’s consideration and included this suggestion.

“Biomass quantification in submerged cultivations was done in a spectrophotometer at 570 nm…”

 Figures

1- Is it possible to separate TPA in a secondary axis for a better view?

R: We acknowledge reviewer’s consideration and inform that we have altered the graphs that show the use of biocatalysts for SSF in the hydrolysis reactions of BHET and PC-PET (Images 1 (c) and (d)). We did not change those related to CALB, because in the case of the hydrolysis of BHET (a), high conversions were observed from 1 hour onwards and the one that presented hydrolysis of PC-PET (b), the formation of intermediates was not observed.

- 7: Same as figure 1.

R: We appreciate the suggestion, but we inform you that we did not change the graphs in Figure 7. This is because in submerged cultures, a very low concentration of hydrolysis intermediates was observed (in none of the analyzed times greater than 5 µmol/L), in the case of BHET, often 0. Changing the axis would make the graph very polluted and would impair the visualization of the main result of the figure, which is the formation of TPA throughout the submerged cultivation.

  1. Please, improve resolution.

R: We acknowledge reviewer’s consideration and improved resolution of figure 8.

  1. Please, improve resolution.

R: We acknowledge reviewer’s consideration and improved resolution of figure 12.

Conclusions

Authors mentioned in the introduction: “One of the great appeals of the SSF, which is characterized by the absence or near absence of free water in the system, is linked to the possibility of using residues and/or by-products from the agroindustry as a cultivation medium…”

Given the results, from the authors’ perspective, is SSF still feasible? could it be improved? is SSF still worthy of study for this type of bioprocess?

R: Based on the results of the present work, a better biodepolymerization performance is observed in submerged cultures. This may be related to several factors, such as adhesion, production of other biomolecules that help in the process.

Regarding the development of bioprocesses for enzymatic hydrolysis of PET, solid-state fermentation can help to minimize costs related to the production of biocatalysts. As an example of a possible process to be studied, it would be the expression of some PET hydrolase in Yarrowia lipolytica and its production in SSF.

Such process would have two major advantages, the intense secretory activity of Yarrowia lipolytica and the minimization of production costs. On the other hand, the use of wild Yarrowia strains, in our opinion, should go the way of submerged cultivation.

 References

Please update references (2022).

R: We appreciate the suggestion and agree with it. Our introduction mostly presents recent references on the subject. We understand that some references used in the discussion are a little old; however, they are essential and unique to support our results. This also includes references to materials and methods.

Round 2

Reviewer 1 Report

The editorial team would be able to suggest, the reduction of the number of tables and figures. 

I have no further concern.

Author Response

Rio de Janeiro, October 29th, 2022.

Dear Ramona Li, Assistant Editor,

Molecules,

Thank you for your e-mail dated 25th October 2022 regarding the article entitled “Post-consumer poly(ethylene terephthalate) (PET) depolymerization by Yarrowia lipolytica: A comparison between hydrolysis using cell-free enzymatic extracts and microbial submerged cultivation” (molecules-1950263) and for reviewer’s comments.

Kindly find enclosed a revised version taking into consideration that reviewer’s comments have been very seriously considered and answered. Note: the new changes are included in red in the revised manuscript.

In the last round of review, one of the reviewers suggested to the editors a reduction in the number of figures and tables. However, we consider the presence of such data in the manuscript as essential for the understanding and support of the hypotheses described there. In addition, all data are extensively discussed and supported by other literature data, which justifies their maintenance in the main text.

Let me close by thanking you in advance for the time spent dealing with this manuscript. We look forward to hearing from you in the near future.

Yours sincerely,

Julio Cesar Soares Sales, MSc.

Biochemistry Department

Chemistry Institute

Federal University of Rio de Janeiro

Reviewer 1:

Comments and Suggestions for Authors:

“The editorial team would be able to suggest, the reduction of the number of tables and figures. 

 I have no further concern.”

R: We acknowledge reviewer’s consideration. However, we consider the presence of such data in the manuscript as essential for the understanding and support of the hypotheses described there. In addition, all data are extensively discussed and supported by other literature data, which justifies their maintenance in the main text. Anyway, we presented the proposal to the editors in the cover letter and explained our point of view regarding the maintenance of data in the manuscript.
